# Bioaccessibility and *In Vitro* Intestinal Permeability of a Recombinant Lectin from Tepary Bean (*Phaseolus acutifolius*) Using the Everted Intestine Assay

**DOI:** 10.3390/ijms22031049

**Published:** 2021-01-21

**Authors:** Lineth Juliana Vega-Rojas, Ivan Luzardo-Ocampo, Juan Mosqueda, Dulce María Palmerín-Carreño, Antonio Escobedo-Reyes, Alejandro Blanco-Labra, Konisgmar Escobar-García, Teresa García-Gasca

**Affiliations:** 1Laboratorio de Biología Celular y Molecular, Facultad de Ciencias Naturales, Universidad Autónoma de Querétaro, Av. de las Ciencias s/n, Juriquilla, Querétaro 76230, Querétaro, Mexico; l.julianavega@gmail.com (L.J.V.-R.); dulce_palmerin@hotmail.com (D.M.P.-C.); konisgmar.escobar@uaq.mx (K.E.-G.); 2Programa de Investigación y Posgrado en Ciencias de los Alimentos, Facultad de Quimica, Universidad Autónoma de Querétaro, Querétaro 76010, Querétaro, Mexico; ivan.8907@gmail.com; 3Laboratorio de Inmunología y Vacunas, Facultad de Ciencias Naturales, Campus Aeropuerto, Universidad Autónoma de Querétaro, Carretera a Chichimequillas, Ejido Bolaños, Querétaro 76140, Querétaro, Mexico; joel.mosqueda@uaq.mx; 4Centro de Investigación y Asistencia en Tecnología y Diseño del Estado de Jalisco, A.C., Av. Normalistas 800, Col. Colinas de la Normal, Guadalajara 44270, Jalisco, Mexico; aescobedo@ciatej.mx; 5Centro de Investigación y de Estudios Avanzados Unidad Irapuato, Departamento de Biotecnología y, Bioquímica, Irapuato 36821, Guanajuato, Mexico; alejandroblancolabra@gmail.com

**Keywords:** Tepary bean (*Phaseolus acutifolius*), apparent permeability, bioaccessibility, *ex vivo* everted intestine assay, *in vitro* and *ex vivo* gastrointestinal digestion, recombinant lectin

## Abstract

Tepary bean (*Phaseolus acutifolius*) lectins exhibit differential *in vitro* and *in vivo* biological effects, but their gastrointestinal interactions and digestion have not yet been assessed. This work aimed to evaluate the changes of a recombinant Tepary bean lectin (rTBL-1) through an *in vitro* and *ex vivo* gastrointestinal process. A polyclonal antibody was developed to selectively detect rTBL-1 by Western blot (WB) and immunohistochemical analysis. Everted gut sac viability was confirmed until 60 min, where protein bioaccessibility, apparent permeability coefficient, and efflux ratio showed rTBL-1 partial digestion and absorption. Immunoblot assays suggested rTBL-1 internalization, since the lectin was detected in the digestible fraction. The immunohistochemical assay detected rTBL-1 presence at the apical side of the small intestine, potentially due to the interaction with the intestinal cell membrane. The *in silico* interactions between rTBL-1 and some saccharides or derivatives showed high binding affinity to sialic acid (−6.70 kcal/mol) and N-acetylglucosamine (−6.10 kcal/mol). The ultra-high-performance liquid chromatography–electron spray ionization–quantitative time-of-flight coupled to mass spectrometry (UHPLC–ESI–QTOF/MS) analysis showed rTBL-1 presence in the gastric content and the non-digestible fraction after intestinal simulation conditions. The results indicated that rTBL-1 partially resisted the digestive conditions and interacted with the intestinal membrane, whereas its digestion allowed the absorption or internalization of the protein or the derivative peptides. Further purification of digestion samples should be conducted to identify intact rTBL-1 protein and digested peptides to assess their physiological effects.

## 1. Introduction

Some natural lectins have been investigated due to the fact of their value as phytotherapeutic agents providing health benefits [1,2]. Particularly, lectins from the Tepary bean (*Phaseolus acutifolius*) have gained interest due to the fact of its lower toxicity grade than other legume lectins [3]. Tepary bean lectin fraction (TBLF), obtained through molecular weight exclusion chromatography, has shown *in vitro* differential cytotoxic effects on cancer cell lines, where colon cancer cells were the most sensible ones [4]. As such, TBLF induces apoptosis in human HT-29 colorectal cancer cell death and causes cell-cycle changes such as a G_0_/G_1_ phase arrest. Moreover, apoptosis was triggered through caspase 3 activation and p53 phosphorylation [5]. In order to study the effect of TBLF on colon cancer, it was administered by intragastric cannula to rats previously treated with dimethylhydrazine or azoxymethane as colon cancer inducers, where TBLF exhibited early premalignant lesion inhibition, suggesting its potential effect against colon cancer [6]. Alatorre et al. [7] reported TBLF participation in vivo in the activation of the immune system by inducing changes in the lymphocyte–granulocyte ratio, raising the interleukin-6 (IL-6) and nuclear factor kappa light chain enhancer of activated B cells (NF-κB) levels, and increasing the number of lymphoid follicles in the Payer’s patches as well. Nonetheless, the results indicated lectin-induced intestinal atrophy, pancreatic hypertropia, and decreased body weight gain; the animals, however, recovered after a two-week resting period. Furthermore, TBLF can also negatively affect the intestinal permeability, decreasing protein digestibility, and changes in the occludin distribution [8]. 

Due to the low yield, time-consuming, and high cost process to obtain the TBLF, the production of a recombinant lectin (rTBL-1) expressed in yeast (*Pichia pastoris*) was performed in our laboratory [9]. The structural analysis of the whole polypeptide sequence showed a 277 amino acid chain that displayed high identity with lectin A (LA) and lectin B (LB) [10] and with a *P. acutifolius* lectin previously deduced by Mirkov et al. (1994) [11] (accession number: AAA82181.1, https://www.uniprot.org/uniprot/Q40750). The coverage percentage of identity with the rTBL-1 (https://www.rcsb.org/structure/6TT9) was 97.83%, with six amino acid changes in position 17 P changed to A, in position 139 R changed to K, in position 180 GQ changed to VN, in position 228 R changed to S, and in position 239 T changed to S [12]. It is important to note that none of the sequences of LA and LB reported by Torres-Arteaga et al. (2016) [10] displayed the N-terminal peptide, indicating that it was cleaved during the protein processing. rTBL-1 is approximately 2.5 kDa larger than the native lectin, mainly due to the addition of a 6XHis tag (~840.92 Da), the sequence EAEAAA at the N-terminal, derived from Kex2p cleavage of the a-factor (~561 Da), and possibly the presence of bigger glycosidic antennas than those on TBL-1 (~640 Da). The structure and size of these carbohydrates analyzed by glycosidases treatment, SDS-PAGE and periodic acid Schiff staining suggest that both proteins contain exclusively N-linked glycans [9]. Although the rTBL-1 lacks agglutination activity, it exhibits similar cytotoxic effects to the TBLF (unpublished data). The rTBL-1 production process, at this time, resulted in a yield of 316 mg/mL; the rTBL-1 was stable up to 55 °C and pH 8.5 in Tris-HCl buffer. rTBL-1 was purified by nickel affinity chromatography, and SDS-PAGE electrophoresis showed the presence of a single band of ~30 kDa [9]. 

The way lectins act depends on the effects of the gastrointestinal tract, since this system is responsible for digesting molecules and absorbing nutrients and fluids throughout the intestinal membrane, comprising a dynamic and complex process carried out by paracellular and transcellular absorption mechanisms [13]. Both transportation systems involve two main concepts: bioaccessibility and bioavailability. Bioaccessibility refers to the total bioactive compounds capable of exerting a biological effect under different physicochemical conditions [14]. Bioavailability refers to the rate and ability of a bioactive compound to be absorbed and become available at active sites including circulating metabolites in the bloodstream [15]. 

Several techniques have been developed for the determination of bioaccessibility such as the *in vitro* and *ex vivo* use of tissues or cell cultures, artificial membranes, octanol–water partition coefficient (Log P), perfusion in situ, in vivo approaches, and in silico methods measuring the permeability or absorption of drugs or small molecules [16,17,18]. The CaCo-2 cells model is the most widely *in vitro* method used. However, this model exhibits several limitations such as the presence of more cell junctions than those from normal intestinal tissue, lack of crucial cell types within the absorption process (goblet, endocrine, and M cells), its preference for transcellular transport, and overestimation of bioaccessibility values due to the overexpression of glycoprotein P [19]. 

In contrast, *ex vivo* models offer low-cost competitive advantages, such as the inclusion of whole or perfused organs from intestinal sections, allowing pharmacokinetic and pharmacodynamic studies of specific biological compounds [20,21]. This technique also keeps all stages of the gastrointestinal digestion in an integrated manner and has demonstrated efficiency in studying the oral administration of bioactive molecules [13,22]. 

Most of these procedures can be coupled with analytical methods to identify and quantify biologically active metabolites derived from the digestive process such as HPLC [23,24]. Specifically, for legumes, biologically active peptides generated during the gastrointestinal *in vitro* digestion of the common bean (*Phaseolus vulgaris* L.) have also been identified through electrophoresis and mass spectrometry (MS) [25,26]. Similarly, peptides from the digestive process of milk protein have been generated and identified through reverse-phase high-performance liquid chromatography coupled to mass spectrometry (RP–HPLC–MS/MS) [27].

Considering that the most significant adverse effects of the lectins from Tepary bean are related to their resistance in the intestinal tract, this work aimed to assess the bioaccessibility and intestinal permeability of rTBL-1 using an *ex vivo* and *in vitro* gastrointestinal digestion procedure. To the best of our knowledge, this is the first study assessing these digestive parameters in the gastrointestinal dynamics on rTBL-1 and also the identification of rTBL-1 via Western blot, immunohistochemical, and ultra-high-performance liquid chromatography–electron spray ionization–quantitative time-of-flight coupled to mass spectrometry (UHPLC–ESI–QTOF/MS) methods. 

## 2. Results

### 2.1. Assessment of Intestinal Tissue Viability 

The viability assessment after incubation in the everted gut sacs is shown in Figure 1 using glucose and water flux as predictors of intestinal integrity. The quantification of glucose is considered a functional parameter because of the integrity of glucose transporters (SGLT1 or GLUT2) in viable intestinal tissue. When the concentration of glucose is low in the intestinal lumen, only SGLT1 is located in the apical side to transport it. Nonetheless, if the glucose concentration is high, GLUT2 is also located in the apical side to support glucose absorption [28]. Glucose enters from the apical side to the basolateral side of the small intestine, confirming the functionality of the glucose transporters and the gut sacs’ viability [29,30]. However, if GLUT2 is located on the apical side, a low opposite glucose transport is observed [28] (Figure 1A). 

Figure 1B,C show the viability and functional assay for the everted intestine using high glucose concentration (9.25 ± 0.39 at the apical side). The apical glucose concentration decreased (30–90 min), while the basolateral concentration was the opposite, suggesting apical to basolateral glucose transport. 

The water flux is another parameter that confirms the viability of the tissue. Water passes through enterocytes by passive transport (osmosis), co-transporters (membrane proteins serving as channels in the water transference), small proteins (aquaporins), or paracellular transport [31]. Apical to basolateral water flux produces water gain, increasing the gut sac weight. On the contrary, basolateral to apical water flux produces weight loss. Our results showed a decrease in the gut sac weight, indicating basolateral-apical movement that increased between 30 and 60 min with a low reversion at 90 min (Figure 1C).

Representative images of the everted jejunum (Figure 2), stained with a hematoxylin and eosin (H&E) procedure, showed that between 30 and 60 min, the intestinal tissue maintained its structure for both the control and rTBL-1-treated samples. At 90 min, there was increased damage to the intestinal tissue, suggesting the feasibility of using this model for studying the intestinal absorption of food-derived matrices for up to 60 min. Therefore, the subsequent assays were performed until 60 min. 

### 2.2. Bioaccessibility and Intestinal Permeability of rTBL-1

Figure 3 shows the effect of the digestion process on rTBL-1 for the gastric content (GC) and intestinal content (IC), where protein concentration was measured after each step of the *in vitro* digestion for bioaccessibility and permeability determination. The GC showed 67% protein, suggesting that 33% was digested in the stomach. The non-digestible fraction (NDF) (apical side) showed 49% at 30 min and 15% after 60 min (*p* < 0.0001), suggesting that intestinal and pancreatic enzymes digested 52% of the protein after 60 min. The digestible fraction (DF) (basolateral side) showed 45 and 43% of protein content after 30 and 60 min of intestinal digestion, respectively (*p* > 0.05) (Figure 3A), suggesting protein absorption or internalization. 

The apparent permeability coefficient (P_app_) (Figure 3B) showed that the protein permeability rate decreased 2.15 times for the apical–basolateral flux (*p* > 0.05) and 6.4 times for the basolateral—apical flux (*p* < 0.0001) between 30 and 60 min. Additionally, the efflux ratio (ER) decreased between 30 and 60 min of intestinal incubation (*p* > 0.05). 

### 2.3. Electrophoretic Profile and Western Blot Analysis of rTBL-1 through the In Vitro and Ex Vivo Digestion

In order to identify the rTBL-1, an anti-rTBL.1 antibody was produced in rabbits. Figure 4 shows the antibody titles after three immunizations with rTBL-1. It is observed that the pre-immune serum (purple diamond) was below the threshold value of 0.8. The first immunization testing (21 days post first immunization) started with an optical density (OD) of 0.15, and it decreased as longer serum dilutions were assayed, being positive up to a dilution 1:1600. For the second (pink square) and third (green triangle) immunizations tests (15 and 10 days post-immunization), OD was higher than 0.15, exhibiting the same trend as the first immunization but reaching longer dilutions above the threshold value (1:12,800 and 1:25,600, respectively).

The rTBL-1 was monitored using SDS-PAGE electrophoresis at all stages of the gastrointestinal process. Figure 5 shows the electrophoretic and Western blot profiles of the NDF and the DF during the *in vitro* and *ex vivo* gastrointestinal digestion. The rTBL-1 has an apparent molecular weight of 30 kDa, observed in the GC, indicating its resistance to these conditions. Enzyme bands, such as pepsin (37 kDa, blue star) at GC, and several bands corresponding to the digested products from pancreatin at the intestinal digestion (NDF and DF) were also detected. However, bands at ~24, 18, and 13 kDa were identified only in the digested sample protein in NDF and DF (blue boxes). Since faint bands were also found close to ~30 kDa throughout the intestinal stage of the digested sample, an additional Western blot analysis was conducted for both gels to confirm the presence of rTBL-1. As expected, no bands were shown in the control. The rTBL-1 bands were present in the GC and the NDF of the digested sample protein. Low intense bands were also observed in the DF (blue arrows), suggesting a potential internalization of the rTBL-1.

### 2.4. UHPLC-ESI-QTOF/MS Analysis rTBL-1 and Digested Samples 

Figure 6 shows the mass spectra of the rTBL-1. The total ion current (TIC) presents several peaks at different retention times (2.68, 14.63, 19.78, 21.02, and 25.17 min) (Figure 6A). The peak at 2.68 min corresponded to a protein that was analyzed by mass spectroscopy (Figure 6B) with a major peak at 917.9 *m*/*z*. The deconvolution spectrum of this peak is shown in Figure 6C with a single peak with a molecular mass of 29,340.6016 Da.

The mass spectrometry analysis of GC, NDF, and rTBL-1 is shown in Figure 7. TIC analyses for GC, and NDF at 30 and 60 min samples showed a peak close to the same retention time corresponding to the rTBL-1 peak (2.68 min). Nevertheless, the mass spectra of all digested samples in the 0–1175 *m*/*z* range showed more than one protein at the retention time of 2.761 min. The blue boxes in the GC and 30 min NDF images showed a lower intensity protein profile. The deconvolution spectra for this zone for GC (red box) showed several peaks matching with pepsin isoforms (36,762 and 34,966 Da). An approach was taken in this zone within a 0–923 *m*/*z* range, where the rTBL-1 peaks were detected (916.8, 917.3, 917.9 *m*/*z*). The same peaks were detected for the GC (916.8, 917.3, and 917.8 *m*/*z*), NDF at 30 min (916.8, 917.3, and 917.9 *m*/*z*), and NDF at 60 min (916.7, 917.3, and 917.7 *m*/*z*). It was not possible to show the deconvolution spectra of the extended zones because not enough peaks were detected for this process. Due to the detection limit of the instrument (100 µg/mL pure protein), and the low amount of intact protein at the digestible fraction (DF), it was not possible to obtain results. 

### 2.5. Immunohistochemical Analysis

Figure 8 shows representative immunohistochemical images for rTBL-1 in the everted gut sac (jejunum) using the anti-rTBL-1 polyclonal antibody at 30 and 60 min of the intestinal digestion. Unlike the control, a brown signal was observed in the epithelial cells for the rTBL-1-treated samples. Amplified images (63×) are shown to improve the visualization of the rTBL-1 signal (small insets in Figure 8).

### 2.6. In Silico Approach of rTBL-1 Interaction with Small Intestine Carbohydrates or Derivatives 

Table 1 and Figure 9 show the potential in silico interactions between rTBL-1 and representative small intestine carbohydrates or carbohydrate derivatives such as β-D-mannose (Figure 9A), N-acetyl-β-D glucosamine (Figure 9B), sialic acid (Figure 9C), N-acetyl-galactosamine (Figure 9D), β-D-galactose (Figure 9E), α-D-glucose (Figure 9F), and α-L-Fucose (Figure 9G).

Sialic acid and N-acetyl-β-D glucosamine exhibited the highest binding affinity as indicated by the lowest binding energies (–6.70 and –6.10 kcal/mol). In contrast, both mannose and glucose showed the lowest interaction (–5.40 kcal/mol). All ligands showed conventional hydrogen bond binding, but N-acetyl-galactosamine also showed additional carbon–hydrogen bonds and unfavorable interactions, and α-D-glucose exhibited pi–sigma interaction, carbon—hydrogen bonds, and van der Waals forces.

## 3. Discussion

Tepary bean lectins are valuable proteins that have exhibited several biological effects such as differential in vitro cytotoxicity on cancer cells by apoptosis induction through caspase activity and in vivo early tumorigenesis inhibition related with p53, the Akt pathway, and caspase-3 activity [6]. However, adverse intestinal effects were observed, such as atrophy, affected integrity, and decreased protein digestibility [7,8], mainly due to the ability of lectins to act as toxic allergens and hemagglutinins [32]. Such effects indicate the need for further research regarding the dynamics of their gastrointestinal digestion and interaction in order to understand their absorptive potential and their ability to exert the expected biological effects on target organs. 

Lectins can be monitored using suitable techniques to detect whole protein or peptides in tissues and fluids [33,34,35]. For instance, pharmacokinetics parameters of ABNOVA VISCUM Fraxini^®^, a mistletoe (*Viscum album* L.) lectin-containing anticancer drug (20 µg/mL), were calculated using a sandwich enzyme-linked immunosorbent assay (ELISA) coupled to polymerase chain reaction (PCR) amplification [36]. Similarly, *Griffithsia* sp. lectins were monitored in serum and in feces using ELISA [37]. Polyclonal antibodies for detecting lectins from *Sclerotium rolfsii* have been developed, validating its results after serum and organ evaluation through indirect ELISA [38]. Previously, the presence of Tepary bean lectins (TBLF) was determined by Western blot using an anti-phytohemagglutinin antibody from *Phaseolus vulgaris* (Cat. No. AS-2300, Vector Laboratories Inc., Burlingame, CA, USA) was used for the identification of Tepary bean lectins by western blot, with low specificity [3]. Later on, a polyclonal antibody against TBLF was developed. Recently, as it was necessary to produce a recombinant lectin from Tepary bean (rTBL-1) by using *Pichia pastoris* yeast [9], we developed a specific polyclonal antibody against the recombinant lectin for its detection in specific tissues, serum, and fluids throughout the gastrointestinal tract. 

It is known that lectins are proteins highly resistant to digestion because they can bind to intestinal epithelial cells receptors depending on lectin’s size, charge, the presence of selected functional groups, the cell glycosylation pattern, and the secondary and tertiary protein structures [34,39]. Furthermore, lectins resist bacterial degradation [40], surviving the digestive tract conditions and remain intact in their biological and immunological form [39]. On an in vivo assay, 78% of an intragastrically administered lectin from Kintoki beans (*Phaseolus vulgaris* variety Kintoki) was found in the gastrointestinal tract after 24 h [41]. Additionally, up to 90% of Concanavalin A was recovered in rat feces after a 4 h post-oral administration, and the lectin was still detectable for 4 days [42]; and orally administered red algae (*Griffithsia* sp.) lectins were resistant to enzymatic degradation as they were recovered after 9 days in feces [37]. In the case of lectins from TBLF, we observed that they resisted the digestive tract conditions after an intragastric administration to Sprague–Dawley rats, indirectly detected at 72 h in feces by agglutination activity [3].

In vitro gastrointestinal digestion has been used to evaluate the digestive process for some food and food-derived matrices including protein samples [43]. The use of the everted gut sac technique allows for the study of the interaction of food components where intestinal viability and integrity are preserved for several minutes to hours. Here, we determined that glucose and water transportation assays confirmed the everted gut sac viability until 90 min; however, although the low reversion of the everted gut sac weight at 90 min was into the same range, it may suggest a loss of intestinal integrity. The histological assays confirmed that intestinal integrity was affected after 90 min of incubation, therefore, the functionality of the tissue was guaranteed until 60 min. 

Protein bioaccessibility was assayed by determining the ratio of protein concentration in each step of the gastrointestinal digestion. It was observed that rTBL-1 bioaccessibility decreased in the intestinal lumen (NDF) and increased in the basolateral side (DF), suggesting a partial digestion of rTBL-1 that may include the absorption of some peptides, or the internalization of the whole protein. Meanwhile, the apparent permeability coefficients indicated that the rTBL-1 was partially absorbed or internalized. The values for the apparent permeability allow the prediction of low (P_app_ < 1 × 10^−7^ cm/s) and high (P_app_ > 1 × 10^−6^ cm/s) permeation, based on the original CaCo-2 cells permeation kinetics [44]. Although no significant difference was observed, the apparent patency coefficient for apical-basolateral flux decreased 2.5 times. However, a significant decrease, 6.4 times, was observed for the basolateral-apical flux. Based on these values, rTBL-1 provoke high membrane permeability, agreeing with the enhanced permeability effect derived from lectins such as crude red kidney beans (*Phaseolus vulgaris* L.) lectins [45]. Regarding the efflux ratio, it is known that a value between 1.5 and 2.0 means that the flux rate is concentration dependent and saturable; conversely an ER < 1.5 is carried out by non-saturable pathways [17]. Here, we found that the ER decreased from 30 to 60 min suggesting a non-saturable mechanism. It has been shown that most peptides and proteins can be absorbed by the endocytosis mechanism [18]. Nevertheless, it has been suggested the involvement of P-glycoprotein (P-gp), an efflux transporter, on the in vitro bidirectional transporting of drugs, proteins, and peptides from the apical to basolateral side and vice versa [46,47]. Particularly for amino acids and dipeptides, there are other bidirectional transporters such as GLYT1, CAT1, SNAT2, PEPT-1, and PEPT-2, dependent on the transmembrane proton gradient, the luminal amino acid concentration, and the cellular traffic [20,48]. 

The protein profile of the digested samples was studied by SDS-PAGE, Western blot, and immunohistochemical analysis in the gastric and intestinal contents. Differential protein bands were observed in both the NDF and the DF, suggesting a digestive process and a potential absorption. The internalization of the whole protein could be possible since it was detected in the DF by Western blot. Additionally, the mass spectrometry analyses agree with the presence of the rTBL-1 in low concentrations in the gastric content and in the NDF, which suggests the protein’s resistance to the digestion process. However, it was not possible to determine the residual peptides because the high protein content from the digestive enzymes. This result agrees with the gastric digestion resistance value (INFOGEST-simulated digestion procedure) of phytohemagglutinin and phaseolin proteins from black beans (*P. vulgaris* L.), determined by SDS-PAGE electrophoresis and HPLC-MS detection with a partial degradation at the intestinal stage [49]. Dimitrijevic et al. [50] also reported intact lectins from banana after the gastric digestion but gradual degradation at the intestinal stage.

During intestinal digestion, lectins can be linked to free carbohydrates or glycoconjugates of cell membranes by a carbohydrate recognition domain (CRD), formed by conserved residues impacting their conformation and function [51]. The CRD selectivity is modified by the possibility of interacting through hydrogen bonds with hydroxyl groups from carbohydrates and van der Waals forces [52]. It was possible to detect the rTBL-1 by immunoassay in the apical side of the intestinal tissue, suggesting an interaction between the lectin and the cell surface. Among the main glycoconjugates, N-acetyl-galactosamine, N-acetyl-glucosamine, galactose, fucose, and sialic acid comprise 77.5% of the intestinal mucus that forms a continuous and protective layer with mucins. Differences in the glycosylation pattern represent a wide variety of potential binding sites for the intestinal absorption of lectins [53]. To confirm the potential interactions between rTBL-1 and the small intestine receptors, in silico interactions were determined. The rTBL-1 was subjected to an additional in silico digestion, but none of the generated peptides were big enough to conduct an interaction (peptide mass < 500 Da, data not shown). Previous studies using glycan arrays and isothermal titration calorimetry showed that rTBL-1 recognized 14 β1-6 branched N-glycans independently of their size, where β-D-galactose, β-D-N-acetyl-glucosamine, β-D-mannose, α-L-fucose, and sialic acid were found to be part of the recognized glycans [9]. Sialic acid and N-acetyl-glucosamine exhibited the highest binding affinity, as indicated by the lowest binding energies, suggesting that the rTBL-1 could directly interact with membrane components and its possible internalization as shown by Western blot, immunohistochemistry, and mass spectrometry analyses. It has been shown that rich-proline peptides resist the digestive enzymes from the intestinal brush borders (peptidases) in vitro, contributing to its crossing through the intestinal barrier [25,54]. Glycosylated receptors also support the intestinal absorption of lectins. For instance, N-acetyl-glucosamine or asialofetuin receptors contribute to Concanavalin A and wheat germ agglutinin intestinal recognition followed by internalization by cellular endocytosis; however, lectins exhibited a non-selective binding pattern to the epithelium. It has been shown that some lectins attach to mannose in jejunum epithelial cells [53]. Once the available lectin receptors are saturated, the remaining lectin is eliminated steadily in a non-linear process [55]. Some in vitro assays showed lectins’ ability to irreversibly bind to epithelial cells’ surface due to the carbohydrate-based building blocks [45].

## 4. Materials and Methods

### 4.1. Production of rTBL-1

*Pichia pastoris* cells were grown on minimal glycerol (MD) plates containing 1.34% (*w*/*v*) yeast nitrogen base without amino acids (YNB), 4 × 10^−5^% (*w*/*v*) biotin, 2% (*w*/*v*) glycerol, and 2% (*w*/*v*) zeocin agar and stored at 4 °C. Long-term stocks were prepared as recommended by Invitrogen (*Pichia* Expression Kit, Version M) and stored at –80 °C. For rTBL-1 production, a single *Pichia pastoris* (SMD1168H strain) colony was used to inoculate 5 mL Yeast–Peptone–Glycerol (YPG) extract: 1% (*w*/*v*) yeast extract, 2% (*w*/*v*) peptone, 2% (*w*/*v*) glycerol and 4 × 10^−5^% (*w*/*v*) biotin. The culture was grown overnight at 30 °C. A 0.5 L baffled flask was inoculated with 0.5 mL of the overnight culture and incubated at 30 °C to generate the inoculum for the bioreactor. Fermentations were carried out in a 2.1 L working volume using an In-Control (Applikon, Delft, The Netherlands) interfaced with Lucullus PIMS Lite software version 3.7.2 (Applikon) for data acquisition and supervisory control. A starting volume of 1.1 L of modified basal salts medium ((MSM), 0.93 g/L CaSO_4_·2H_2_O, 18.2 g/L K_2_SO_4_, 14.9 g/L MgSO_4_·7H_2_O, 4.13 g/L KOH, and 26.7 mL/L H_3_PO_4_, 40 g/L glycerol), and 0.5 mL antifoam (VRF-30, Sigma–Aldrich, St. Louis, MO, US) was sterilized inside the reactor. Ammonium hydroxide (15%, *v*/*v*) was used as a pH control agent and nitrogen source (pH 5.0). PTM1 trace salts (24 mM CuSO_4_, 0.53 mM NaI, 19.87 mM MnSO_4_, 0.83 mM Na_2_MnO_4_, 0.32 mM boric acid, 2.1 mM CoCl_2_, 0.15 mM ZnCl_2_, 0.23 M FeSO_4_, and 0.82 mM biotin) were aseptically added at 4.35 mL/L after sterilization prior to inoculation. The process was switched to 0.5 L fed batch using glycerol–MSM 1:1 added to the bioreactor with constant feeding flow rate (F) of 0.06 mL/min. After the fed-batch system, the fermentation supernatant was clarified through a 0.22 µm membrane filter and purified from the supernatant by nickel-affinity chromatography using HisTrap Hig performance columns. The resultant protein was dialyzed, lyophilized, and stored at −20 °C [9].

### 4.2. rTBL-1 Antibody Design 

First, two New Zealand male rabbits (9–10 weeks age) were immunized. During the adaptive and experimental period, the animals were placed in boxes with water and food (Conejina T, Purina, St. Louis, MO, USA) *ad libitum* with a 12:12 light–dark circadian cycle at 21 ± 2 °C and relative humidity of 60 ± 5%. The animals were inoculated three times with 1000 µL of the vaccine from a 1 mg/mL stock solution of the freeze-dried recombinant lectin from Tepary bean (rTBL-1) in phosphate buffer solution (PBS) 1×, pH 7.4. For the inoculum preparation, 100 µL of the stock solution was mixed with 400 μL PBS and 500 μL Montanide 71G adjuvant (Seppic, Paris, France) over 15 and 21 days. Pre-immune and post-immune bleeding of the auricular artery was conducted, and the blood was centrifuged (1500× *g* for 15 min) to separate the serum, and subsequently stored at –80 °C. 

The evaluation of antibody titers was done by indirect ELISA where the 96-well flat-bottom polystyrene plates were coated with 100 µL of rTBL-1 overnight at 4 °C. Previously, rTBL-1 (10 μg/mL) was prepared in 0.1 M carbonate and bicarbonate buffer (pH 9.6). Then, the plates were washed 3 times with PBS 1×, supplemented with 0.05% Tween-20 (*v*/*v*) (PBST) at room temperature (25 ± 1 °C) to discard all the unbound peptides. The plates were blocked with 200 µL/well of PBS pH 7.4, 0.05% Tween-20 (*v*/*v*), and 5% low-fat milk, and were incubated at 37 °C for 1 h. Afterward, the plates were washed 3 times, as mentioned above, and tapped down on clean blotting paper. Then, 1:100 proportions of each tested sera (primary antibody, 100 μL/well) were added, and serial dilutions in triplicates were loaded and incubated (37 °C, 1 h). The unbounded antibodies were eliminated by washing twice with PBST 0.05%. Horseradish peroxidase-conjugated anti-rabbit IgG, H+L (111-035-003 Jackson ImmuneResearch, Baltimore, MD, USA) was used as the secondary antibody, diluted at 1:10,000 with PBS 1× and 2% skim milk, and incubated (37 °C, 1 h). Then, the plates were washed three times to eliminate unbound antibodies. The plates were revealed with 0.4 mg/mL of *O*-phenylenediamine dihydrochloride (Sigma–Aldrich, St. Louis, MO, USA) and 4 μL H_2_O_2_ (30% *v*/*v*) in citrate and citric acid buffer. Readings of the plates were carried out in an ELISA plate reader (Microplate Absorbance Reader, Bio Rad, Hercules, CA, USA) at 450 nm, 20 min after adding the revealing solution. The MPM 6.exe software (Bio Rad) was used to analyze the readings. Each sample was analyzed in triplicates, and the limit was calculated with the negative samples, considering the mean ± 3SD. All the samples below this value were considered as negative. 

### 4.3. In Vitro and Ex Vivo Gastrointestinal Digestion 

The human gastrointestinal digestion simulation was conducted following the methodology reported by Campos-Vega et al. [22]. The digestive process was mimicked from the gastric to intestinal digestion, considering that the delivery of lectins for therapeutic purposes involves an intragastric administration, avoiding oral contact. A rTBL-1 solution (5 mg/mL) was dissolved in a pepsin solution (0.055 g in 0.94 mL of 20 mM HCl) (≥2500 U/mg protein, Sigma–Aldrich, St. Louis, MO, USA) and adjusted to pH 2.0, followed by incubation (37 °C, 2 h). The intestinal phase was simulated by adjusting the gastric fraction to pH 7.2–7.4 and adding a mixture of intestinal enzymes: 2.6 mg pancreatin (8 × USP, Sigma–Aldrich, St. Louis, MO, US) and 3.0 mg bile bovine (Sigma–Aldrich), dissolved in 5 mL Krebs–Ringer buffer solution (pH 6.8, CO_2_-gasified, 37 °C) (118 mM NaCl, 4.7 mM KCl, 1.2 mM MgSO_4_, 1.2 mM KH_2_PO_4_, 25 mM NaHCO_3_, 11 mM glucose, and 2.5 mM CaCl_2_, pH 6.8). 

The intestinal mixture was added to a test tube, and an everted gut sac was placed in the solution. The sac was excised from male Sprague–Dawley rats (250–300 g body weight, 6–8 weeks age) fasted 16 h before the procedure with water *ad libitum* and euthanized by decapitation. The gut sac (jejunum) was cut into 6 cm segments, carefully everted, tied by one side, and filled with the Krebs–Ringer solution (37 °C), before an additional ligature. The everted sacs were incubated in the intestinal solution for 30 and 60 min using an oscillating water bath (37 °C, 80 cycles/min). Aliquots from outside (apical side or non-digestible fraction, NDF) and inside (basolateral side or digestible fraction, DF) the everted sacs were collected and stored at –80 °C for further ultra-high-performance liquid chromatography-mass spectrometry (UHPLC-MS) analysis. A control prepared with distilled water was subjected to the same procedure. Three independent experiments in triplicates were conducted. The Bioethics Committee of the Department of Natural Sciences from Universidad Autónoma de Querétaro approved the experimental procedure (44FCN2015). The procedure complied with the National Institute of Health (NIH) guide for the care and use of Laboratory Animals. An overall scheme of the followed procedure is shown in Figure 10. 

### 4.4. Assessment of Intestinal Tissue Viability

#### 4.4.1. Histopathology Analysis 

For the histopathological evaluation, the jejunum was fixed in 10% formaldehyde, treated by alcohol-series dehydration, and stained with hematoxylin–eosin for subsequent histological analysis by microscopy (DM2500 model, Leica, Wetzlar, Germany) with 10× and 63× lens.

#### 4.4.2. Water Flux and Glucose Transport

To assess the small intestine tissue viability, the everted jejunum was incubated in Krebs–Ringer buffer. The water flux (WF) was calculated according to the equation: WF = (W_3_ − W_2_)/W_1_, considering WF as the water flux (g water/g fresh intestine), where W_1_ is the initial small intestine weight (without Krebs–Ringer buffer filling), W_2_ is the buffer-filled everted sac weight before incubation time, and W_3_ is the everted sac segment weight after the incubation period [56]. 

The glucose transport was assessed using a saturated glucose solution (4 g/L, dissolved in Krebs–Ringer buffer without glucose) [57]. This solution was pH adjusted (7.2–7.4), mixed with the intestinal enzymes as indicated in Section 4.3. (pancreatin and bile bovine) and subjected to intestinal incubation in the everted gut sac. The apical side contained the glucose solution in the glucose-free Krebs–Ringer buffer and the intestinal enzymes, and the basolateral side contained Krebs–Ringer buffer without glucose. After the incubation (30, 60, and 90 min), aliquots from the inner side and the outer side of the intestine were preserved at 4 °C. The glucose concentration was measured using the Glucose Assay Kit (GAGO20-1KT, Sigma–Aldrich, St. Louis, MO, USA) and following the manufacturer’s instructions.

### 4.5. Effect of Gastric and Intestinal Digestion on rTBL-1

#### 4.5.1. Bioaccessibility Measurement

The amount of rTBL-1 during the *in vitro* and *ex vivo* gastrointestinal digestion was quantified by NanoDrop spectrophotometry at 280 nm (NanoDrop 2000c, Thermo Fisher Scientific). The rTBL-1 was quantified at the start of the experiment (C_0_) and after each step of the in vitro digestion process (treatment groups). A control group was tested without rTBL-1 (control group). In each step of the assay, the difference between treated group minus control group was calculated to obtain the final protein concentration (C_f_) after gastric and intestinal digestion (30 and 60 min). These values were used to calculate the bioaccessibility (B) of protein or the generated peptides after the digestion process, using the equation [58]: B (%) = (C_f_/C_0_) ∗ 100

#### 4.5.2. Apparent Permeability Coefficients (P_app_) and Efflux Ratio (ER)

The P_app_ from the apical to basolateral side (P_app A to B_) and basolateral to apical side (P_app B to A_) was calculated using the equation [17]:Papp: (dQ/dt) × (1/AC_0_)
where P_app_ (cm/s) is the apparent permeability coefficient, dQ/dt (mg/s) is the ratio between the amount of protein or peptide transported across the membrane per unit time, A is the cross intestinal area (cm^2^) that represents the surface area of the everted gut segment available for permeation, and C_0_ (mg/mL) expresses the initial concentration of protein or peptides (gastric content) that are permeating the gut sacs. The efflux ratio (ER) was considered the ratio between P_app B to A_ and P_app A to B_. The mean and the standard values were expressed in 10^−5^ cm/s units.

### 4.6. SDS-PAGE and Western Blot

The presence of rTBL-1 at all digestion stages was identified by Western blot methodology. The electrophoresis was performed at 90 V for 90 min in running buffer (25 mM Tris Base, 200 mM glycine, 3.5 mM SDS, pH 8.3), and the molecular weight marker from pre-stained protein ladder (BG00364, Bio Basic, Markham, ON, Canada) was used. A 13% ((29%) Acrylamide/bisacrylamide (1%)) gel was prepared and run in semi-denaturing conditions. rTBL-1 was visualized following staining with Coomassie Brilliant Blue (CBB-R250, Serva, Heidelberg, Germany). The gel was then transferred to nitrocellulose membrane (Bio Rad) at 4 °C and 15 V for 20 min in Trans-Blot^®^ SD Semi-Dry Electrophoretic Transfer Cell (Bio Rad, USA), using transfer buffer (25 mM Tris base, Glycine 200 mM, Methanol 20%, pH 8.3). The transference was verified with Ponceau red stain (Ponceau at 0.2% *w*/*v*, acetic acid at 5% *v*/*v*). The membrane was washed with PBST 1× (10 mM Tris Base, 150 mM NaCl, Tween-20 0.05%) and then blocked with blocking buffer (TTBS: Tween-20 0.05%, 5% skim milk) overnight at 4 °C under shaking (420 rpm). The membrane was incubated with the primary antibody (polyclonal anti-rTBL-1 antibody) at a 1:30,000 dilution in 1× TTBS with 2% skim milk for 60 min at room temperature under constant agitation, followed by 3 washes (1× TTBS, 15 min each one). The membrane was blocked again with blocking buffer for 1 h to RT under constant agitation. The membrane was incubated with the secondary antibody (anti-rabbit IgG, H+L) conjugated with horseradish peroxidase (Jackson ImmuneResearch, Baltimore, MD, USA) diluted at 1:15,000 in the same conditions. Subsequently, the membrane was rewashed with 1x TTBS. The immunodetection was achieved using Amersham^TM^ ECL^TM^ prime Western blotting detection reagent (GE Healthcare, Little Chalfont, Bucks, UK).

### 4.7. UHPLC-ESI-QTOF/MS Analysis

The rTBL-1 and samples mass spectrometry analyses were performed using an ultra-high-performance liquid chromatograph (UHPLC) (ACQUITY Class *H*, Waters Corporation, Milford, MA, USA) coupled to a high-resolution mass spectrometer with a time of flight and electrospray ion source (ESI-QTOF) (model Xevo G2 QTOF, Waters Corporation, Milford, MA, USA). A C_8_ column (2.1 × 100 mm, 1.7 µm, Waters Corporation, Milford, MA, USA) was conditioned with 100% of the mobile phase with solvent A (30% Acetonitrile, ACN, in 0.1% *v*/*v* formic acid 0.1%) for the detection of the intact protein. Solvent B was 0.1% *v*/*v* formic acid in water. The gradient was set as follows: 30% A at 0 min; 50% A for 5 min, 60% A for 12 min, 95% A for 17 min, 5% A for 22 min, 70% A for 27 min, and finally 30% A for 35 min. The injection volume was 10 µL for the standard solution (1 mg/mL) at 0.3 mL/min flow rate. The temperature of the column was 40 °C. The mass spectrometer was run in a positive electron spray ionization (ESI) mode with 100–2500 *m*/*z* range. The capillary voltage was 2.5 kV, and the instrument resolution was 2000. The MassLynx V4.1 (Waters Corporation, Milford, MA, USA) was used for the acquisition of data. Before the mass spectrometry analysis, the gastrointestinal digestion samples were desalted using Sep-Pak C_18_ plus cartridge (Waters Corporation, Milford, MA, USA). The intact protein was detected using the same previously mentioned solvent gradient, but the injection volume was 20 µL. 

### 4.8. Immunohistochemistry Analysis of rTBL-1

Dehydrated samples of intestinal tissues were embedded in paraffin blocks and cut into 3 µm thick positively charged Thermo Fisher slides (Thermo Fisher Scientific, Waltham, MA, USA) using a microtome and subsequently rehydrated. For this, paraffin was removed from the samples at 60 °C for 20 min, and rehydrated in 100% xylene, a series of alcohol washes (99, 96, and 70%), and distilled water for 10, 10, 5, 5, and 5 min, respectively. The epitopes were unmasked placing the samples in a water bath at 100 °C for 15 min in 0.1 M citric acid solution, pH 6.0. The slides were washed three times (PBS 1×, pH 7.2) and incubated with 3% hydrogen peroxide in PBS 1× (30 min) to remove endogenous peroxidase activity. The non-specific binding sites were blocked using 5% skim milk in PBS-Tween 0.1% for 30 min at RT. Tissues were incubated overnight at 4 °C with the anti-rTBL-1 polyclonal antibody, diluted to 1:10,000. The tissues were incubated with the secondary antibody (anti-rabbit IgG, high and low chains (H+L)) conjugated with horseradish peroxidase (Jackson ImmuneResearch, Baltimore, MD, USA) diluted at 1:5000 for 1 h at RT. The reaction was run using a solution of 0.02 mg of diaminobenzidine in 72 mL of PBS 1× and 120 μL of 30% H_2_O_2_ for 15 min. This reaction produced a sepia-colored precipitate in the immunoreactive cells. Samples were contrasted with Harris hematoxylin and were dehydrated using distilled water, alcohol 70%, alcohol 96%, absolute alcohol, and xylene, and mounted with resin. The samples were analyzed under a microscope DM2500 model (Leica, Wetzlar, Germany) with 10× and 63× lens, and photographs were obtained. 

### 4.9. In Silico Assessment of rTBL-1 Interaction with Small Intestine Ligands

An in silico evaluation was performed to assess potential interactions between digested and non-digested rTBL-1 and small intestine ligands. The reported FASTA sequence of rTBL-1 was downloaded from the Protein Databank (code: 6TT9) [9]. As the first 6 amino acids were excluded from the sequence (EAEAAA), it was 3D modeled using the Swiss Model 2.0 online software [59]. The phytohemagglutinin structure (6TT9.1.A) was selected as rTBL-1 structure for the modeling based on the highest identity. 

For selecting the intestinal ligands that potentially interact with rTBL-1, the reported structures indicated by Martínez-Alarcón et al. [9] were used for the docking methodology. The PubChem database was used to download the 3D structures of β-D-mannose (PubChem CID: 439680), N-acetyl β-D-glucosamine (PubChem CID: 24139), sialic acid (PubChem CID: 906), N-acetyl galactosamine (PubChem CID: 35717), β-D-galactose (PubChem CID: 439353), α-L-Fucose (PubChem CID: 439554), and α-D-glucose (PubChem CID: 79025). If a 3D structure was not reported (e.g., sialic acid), it was modeled using MarvinSketch v. 20.9 (ChemAxon, Budapest, Hungary). The docking procedure [60] was conducted by selecting flexible torsions, hydrogen bonds, and docking calculations provided by AutoDock tools [61]. The visualization of the best docking interactions was done in Discovery Studio Visualizer v. 19.1.0.188287 (Dassault Systèmes, Vélizy-Villacoublay, France). 

### 4.10. Statistical Analysis

Data were expressed as the means ± standard deviation (SD). Overall, two independent experiments with at least three replicates were considered for each measurement of each incubation. The statistical analysis was performed using the JMP v. 8.0 software (SAS Institute, Cary, NC, USA), following one-way analysis of variance (ANOVA) and *post-hoc* Tukey–Kramer’s test for multiple comparisons or t-student test for two samples comparison, where the differences were considered significant at *p* ≤ 0.05. 

## 5. Conclusions

In this study, for the first time, we developed a polyclonal antibody against the recombinant Tepary bean (*Phaseolus acutifolius*) lectin that allowed its selective monitoring through the *in vitro* gastrointestinal digestion using viable *ex vivo* tissues. The values of bioaccessibility of rTBL-1 in gastric content showed a high gastric digestive resistance. However, rTBL-1 intestinal bioaccessibility decreased in the apical side after 60 min of incubation and increased in the basolateral side after 30 min suggesting an absorption process or the protein internalization. The high apparent permeability coefficients of the rTBL-1 during the intestinal incubation suggested that rTBL-1 or derived peptides can cross through the intestinal membrane into the enterocyte. Such results, together with the SDS-PAGE and Western blot assays, suggest either partial absorption or internalization of rTBL-1. The MS analysis provided information about the presence of intact rTBL-1 in the gastric content and in the intestinal apical side (non-digestible fraction); however, the presence of digestive enzymes significantly affected its identification. Moreover, rTBL-1 could potentially interact with intestinal receptors, as indicated by the immunohistochemical analysis and the in silico approach, mainly with sialic acid and N-acetyl glucosamine, two of the most abundant intestinal carbohydrates. Further studies will focus on rTBL-1 protein-glycans interaction by using specific carbohydrates in the study of bioaccessibility, in the purification of samples at each step of the *in vitro* digestion for their identification through MS assay and bioactivity determination, and also to know the rTBL-1 internalization route in order to determine its pharmacokinetic and pharmacodynamic parameters as an anticancer molecule. 

## Figures and Tables

**Figure 1 ijms-22-01049-f001:**
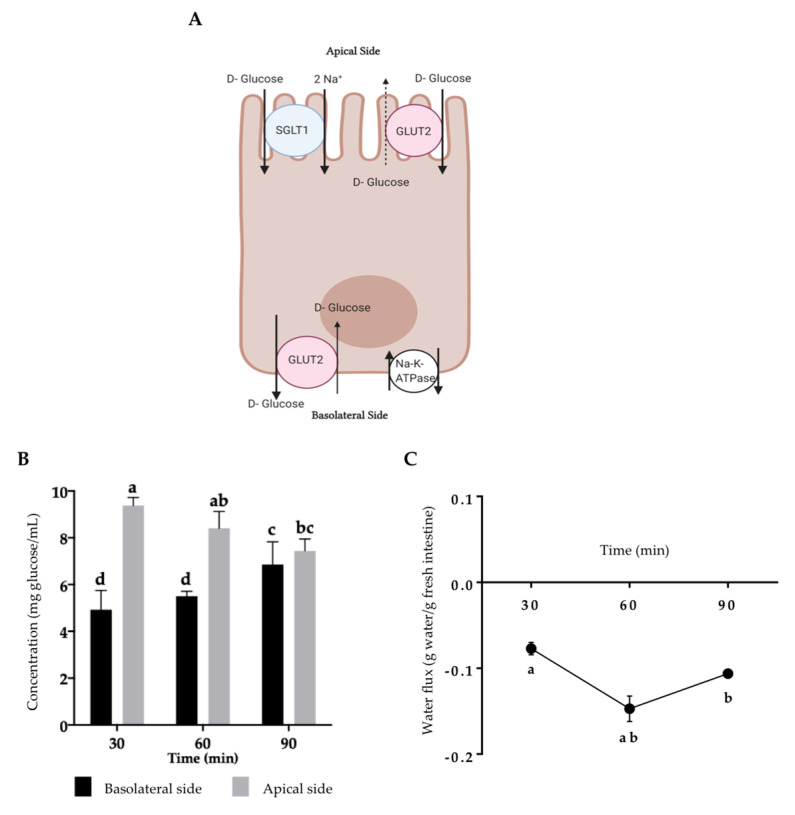
Sprague–Dawley rats’ everted jejunum integrity during small intestine absorption. (**A**) Glucose transportation in enterocytes. (**B**) For the glucose flux determination, everted jejunum was incubated in glucose-free Krebs–Ringer buffer for 30, 60, and 90 min, and glucose was determined at 540 nm. (**C**) For the water flux determination, everted jejunum was incubated in Krebs–Ringer buffer for 30, 60, and 90 min, water was determined as a change in the gut sac weight. Results are expressed as the mean ± SD of three independent experiments. Small letters express significant differences between all the samples (*p* ≤ 0.05) by Tukey–Kramer’s test. The enterocyte model was created with BioRender^®^ (https://app.biorender.com) and adapted from Koepsell [28].

**Figure 2 ijms-22-01049-f002:**
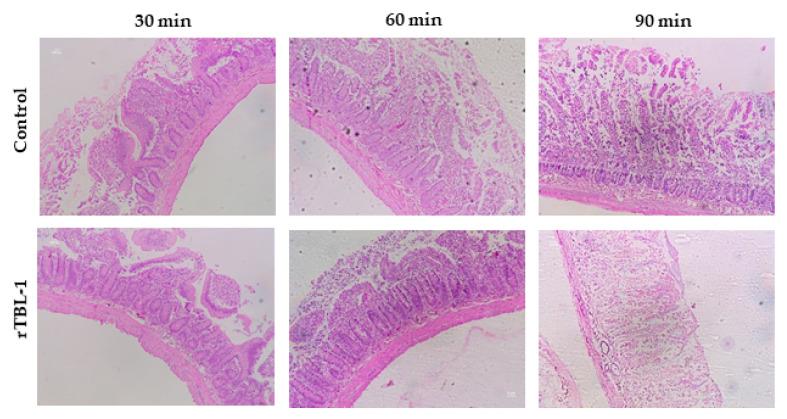
Representative hematoxylin and eosin (H&E) images during the incubation (30, 60, and 90 min). Everted jejunum was fixed in 10% formaldehyde and stained with hematoxylin-eosin for histological analysis (10×). rTBL-1, recombinant Tepary bean (*Phaseolus acutifolius*) lectin. Control corresponded to water (sample without rTBL-1).

**Figure 3 ijms-22-01049-f003:**
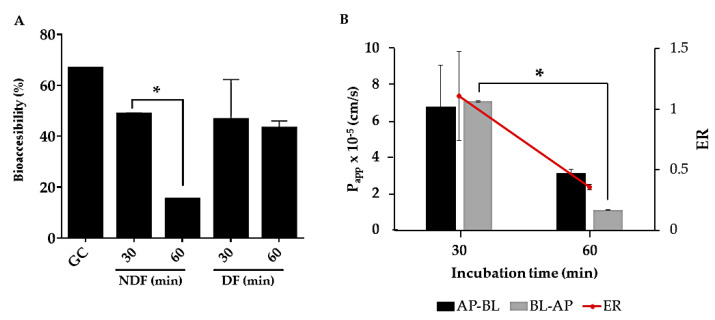
Effects of *in vitro* and *ex vivo* simulated digestion on rTBL-1. Simultaneous gastric and intestinal *in vitro* digestion of rTBL-1 was carried out, and the protein was determined at 30 and 60 min (280 nm). (**A**) Bioaccessibility. (**B**) Apparent permeability coefficient (P_app_) and efflux ratio (ER) during the intestinal incubation. The results are expressed as the mean ± SD of three independent experiments in triplicates. The asterisks express significant differences (*p* ≤ 0.05) by t-student test for each pair of data between 30 and 60 min. AP, apical side; BL, basolateral side; GC, gastric content; NDF, non-digestible fraction; DF, digestible fraction; P_app_, apparent permeability coefficient.

**Figure 4 ijms-22-01049-f004:**
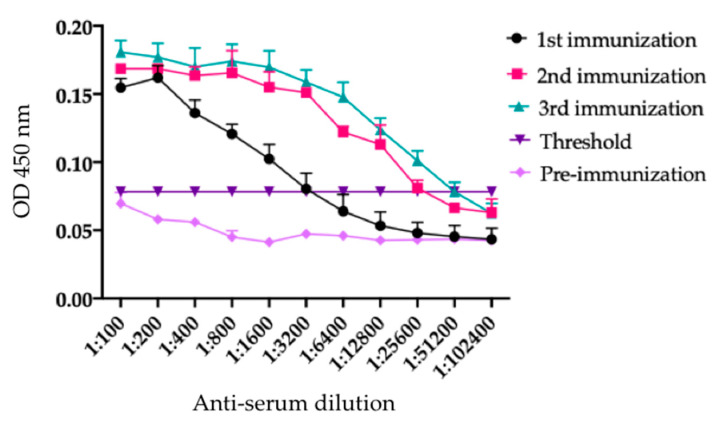
Antibodies titles of rTBL-1. Pre-immune serum corresponded to the period before starting the immunizations. The threshold was calculated with respect to the negative sample means plus three standard deviations. All samples were analyzed for triplicate. OD, optical density.

**Figure 5 ijms-22-01049-f005:**
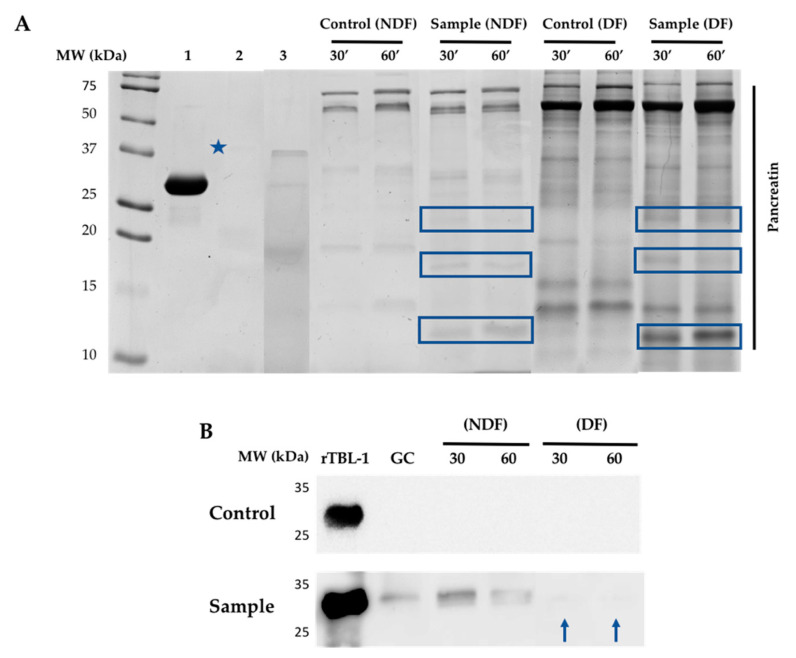
Electrophoretic profile and Western blot for the gastrointestinal effect on rTBL-1. (**A**) 13% SDS-PAGE stained with Coomassie blue. (1) rTBL-1, (2) gastric content (GC) control, (3) GC sample, non-digestible fraction (NDF), digestible fraction (DF). Both NDF and DF were obtained after incubation for 30 and 60 min. (**B**) Western blot for rTBL-1 in digested samples. The columns correspond to the molecular weight (MW) markers, rTBL-1 (2 µg), GC (~60 µg control or sample), NDF (~60 µg control or sample), and DF (~60 µg control or sample). The control was water. The blue star shows the presence of pepsin, blue boxes indicate protein bands derived from the digestion process, and the blue arrows rTBL-1.

**Figure 6 ijms-22-01049-f006:**
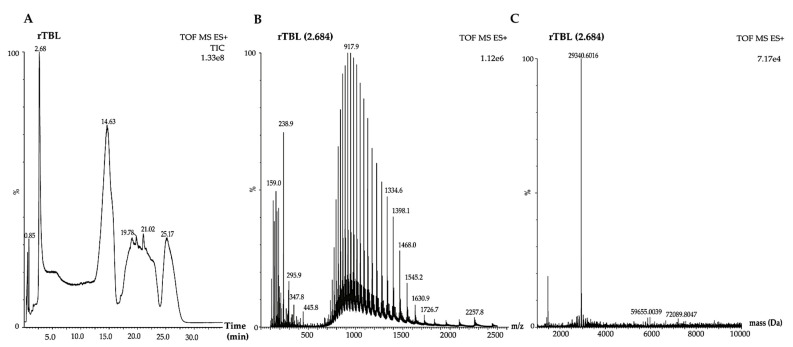
Mass spectrum for rTBL-1. Relative intensity (%) is shown in each graphic. (**A**) rTBL-1 TIC (total ion current). (**B**) Mass spectrum with a range of 100–2500 *m*/*z*. (**C**) Deconvolution spectrum with a range of 10,000:100,000.

**Figure 7 ijms-22-01049-f007:**
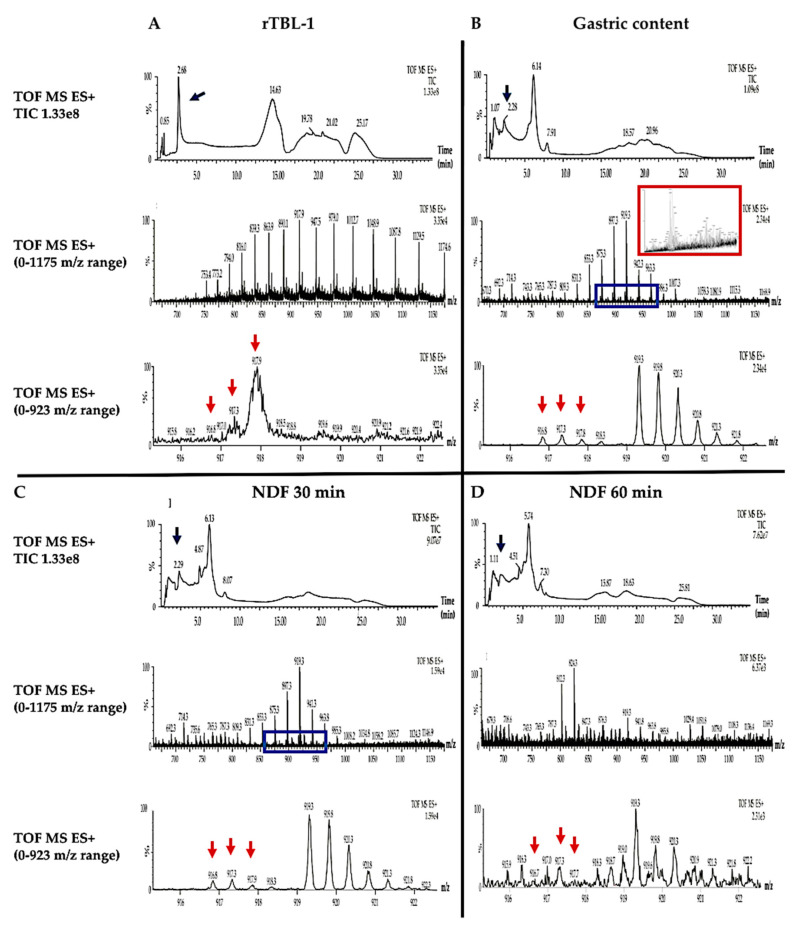
Mass spectrometry for rTBL-1 and samples from the gastrointestinal digestion. (**A**) rTBL-1. (**B**) Gastric content (GC). (**C**) Non-digestible fraction (NDF) 30 min. (**D**) Non-digestible fraction (NDF) 60 min. Blue arrows show the rTBL-1 retention time. The blue boxes show the protein profile of rTBL-1 in the 0–1175 *m*/*z* range. The red box shows the deconvolution spectrum for the gastric content sample. Red arrows show the peaks corresponding to rTBL-1 in the 0–923 *m*/*z* range. Results are shown as the relative intensity (%).

**Figure 8 ijms-22-01049-f008:**
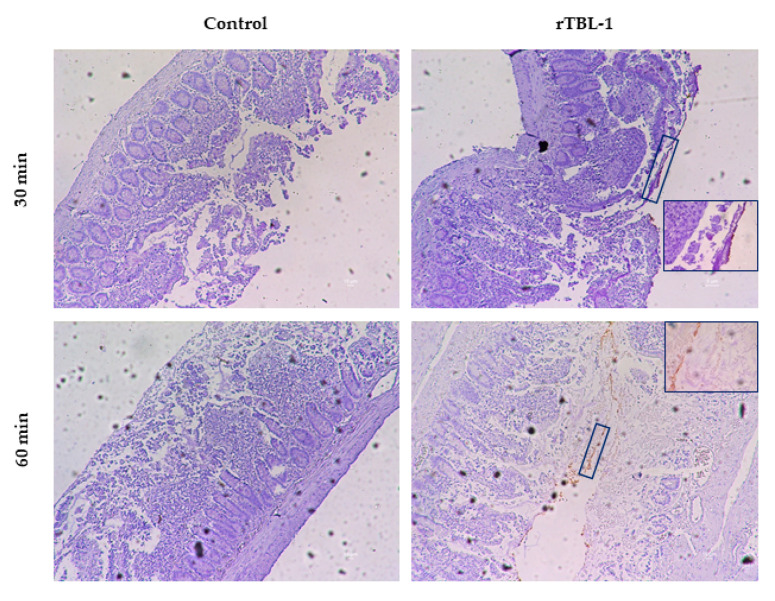
Representative immunohistochemical images of everted rat jejunum. Treated and control samples were assayed for rTBL-1 detection using anti-rTBL-1 polyclonal antibody and contrasted with Harris hematoxylin. Images show 3 µm slices taken at 10× and 63×.

**Figure 9 ijms-22-01049-f009:**
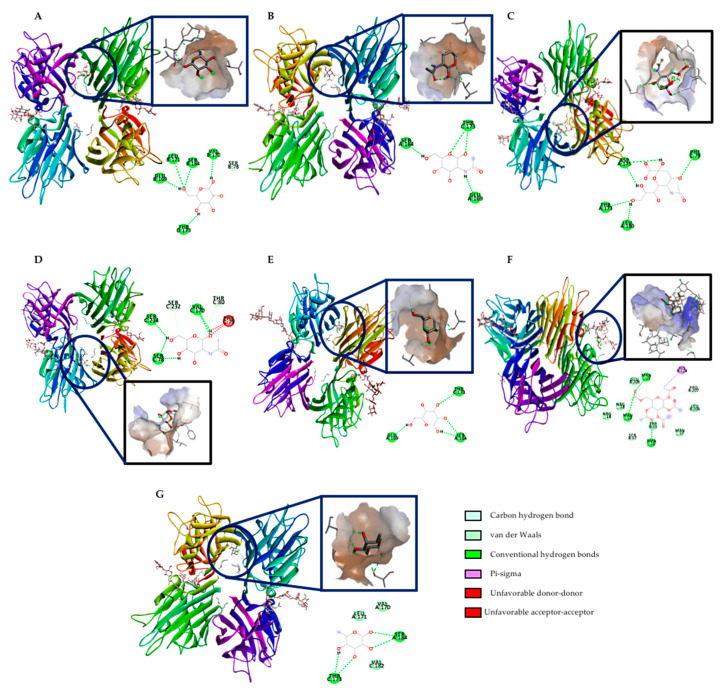
In silico potential interactions between rTBL-1 and intestinal carbohydrates or derivatives. (**A**) β-D-mannose. (**B**) N-acetyl β-D-glucosamine. (**C**) Sialic acid. (**D**) N-acetyl-galactosamine. (**E**) β-D-galactose. (**F**) α-D-glucose. (**G**) α-L-Fucose. The 2D graphic indicates potential chemical interactions with selected amino acids of rTBL-1.

**Figure 10 ijms-22-01049-f010:**
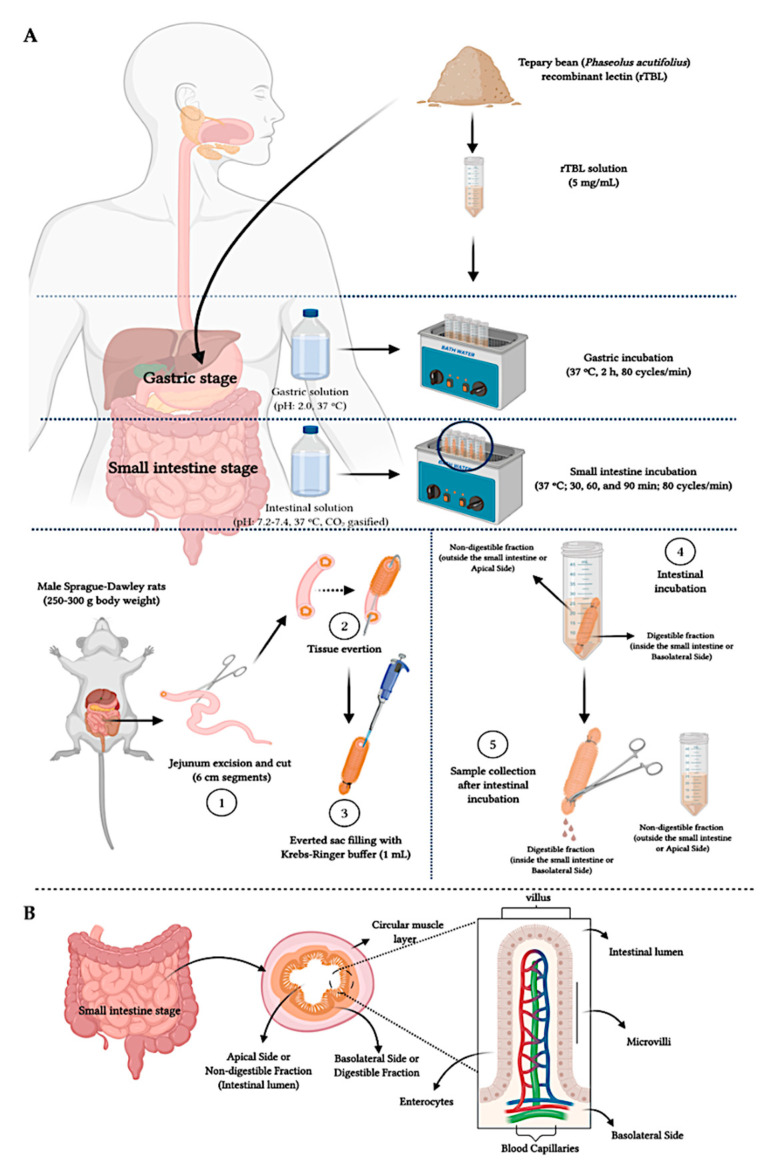
Overall diagram of the in vitro/*ex vivo* gastrointestinal digestion. (**A**) Everted jejunum procedure. (1) The intestinal incubation was simulated *ex vivo* using a jejunum excised from male Sprague–Dawley rats, (2) carefully everted, (3) filled in the inner side (basolateral side) with Krebs–Ringer buffer (1 mL). Once tied on both sides, the filled intestinal tissue was placed in the intestinal solution (4) (pH adjusted sample to 7.2–7.4, added with intestinal enzymes: pancreatin and bile bovine) and incubated (30–60 min). Once incubated (5), the sample from the outer side of the small intestinal tissue was referred to as the non-digestible fraction (apical side), and the inner side of the intestinal tissue was considered as the digestible fraction (basolateral side). (**B**) Jejunum morphology.

**Table 1 ijms-22-01049-t001:** Free energy binding between rTBL-1 and saccharides or derivatives.

Ligands	Binding Energy (kcal/mol)
β-D-Mannose	−5.70
	−5.40
	−5.40
N-acetyl β-D glucosamine	−6.10
	−5.80
	−5.70
Acid sialic	−6.70
	−6.30
	−6.30
N-acetyl galactosamine	−6.10
	−5.80
	−5.60
β-D-Galactose	−5.60
	−5.40
	−5.50
α-D-Glucose	−5.40
	−5.40
	−5.30
	−5.4
α-L-Fucose	−5.4
	−5.4

Data were obtained after molecular docking simulation using AutoDock Vina.

## Data Availability

Data sharing is not applicable to this article.

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
