# Peer review of "Bioaccessibility and In Vitro Intestinal Permeability of a Recombinant Lectin from Tepary Bean (Phaseolus acutifolius) Using the Everted Intestine Assay"

_ijms, 2021, doi:10.3390/ijms22031049_

Round 1

Reviewer 1 Report

Investigation of anticancer properties of lectins is of great interest. Lectins can bind tumor cell bearing specific sugars included in cell membrane. It can be useful for effectiveness of targeted pharmacotherapy of cancer patients.

However, the main problem for lectin is its administration. The manuscript describes the changes of a recombinant Tepary bean lectin (rTBL) through the gastrointestinal process. Authors investigated the process of rTBL digestion in vitro modulating stomach and intestine conditions. They incubated lectin with different enzymes and analyzed the ratio of digested and not digested forms of the protein. Using computer modeling, they tried to predict potential sugar receptors for lectin. The paper is written in a good, accessible language.

The findings are very interesting, experimental strategy is very elegant. I recommend the acceptance of the manuscript after minor considerations.

  • There is no data about sugar specificity of Tepary bean lectin even in previous studies. Sugar specificity is the main characteristic of lectins. I think it will be benefit for manuscript. It will also help to understand the choice of sugars during experiments in silico.
  • In view of carbohydrate specificity, additional experiments should be evaluated to support the idea of protein-carbohydrate interaction. Firstly, bioaccessibility and intestinal permeability of rTBL in the presence of specific glycans. The presence of specific sugars should change the ratio of NDF and DF, because sugars will block the interaction with in intestine cells leading to decreased permeability. Secondly, immunohistochemical analysis in the presence of specific carbohydrates will support the idea of protein-carbohydrate interaction.
  • I think it will be benefit if authors present data on comparison of TBLF and rTBL properties especially in view of cytotoxic effects on cancer cell lines.
  • In my opinion, it will be more harmonic to move data and figures 9 and 10 about antibody design and gastrointestinal digestion from “Materials and Methods” section to “Results” section.

Author Response

Comments and Suggestions for Authors

Investigation of anticancer properties of lectins is of great interest. Lectins can bind tumor cell bearing specific sugars included in cell membrane. It can be useful for effectiveness of targeted pharmacotherapy of cancer patients.

However, the main problem for lectin is its administration. The manuscript describes the changes of a recombinant Tepary bean lectin (rTBL) through the gastrointestinal process. Authors investigated the process of rTBL digestion in vitro modulating stomach and intestine conditions. They incubated lectin with different enzymes and analyzed the ratio of digested and not digested forms of the protein. Using computer modeling, they tried to predict potential sugar receptors for lectin. The paper is written in a good, accessible language.

The findings are very interesting, experimental strategy is very elegant. I recommend the acceptance of the manuscript after minor considerations.

  • There is no data about sugar specificity of Tepary bean lectin even in previous studies. Sugar specificity is the main characteristic of lectins. I think it will be benefit for manuscript. It will also help to understand the choice of sugars during experiments in silico.

Thank you for your comment. Glycan array and ITC analyses were done previously (Martínez-Alarcón et al. Biomolecules 2020,doi:10.3390/biom10040654) and 14 branched glycans from 585 analyzed were identified, where b-D-galactose, b-D-N-acetyl-glucosamine, b-D-mannose, a-L-fucose and sialic acid were found to be part of the recognized glycans. Text was added in page 12, lines 369-372 and 379-382.

  • In view of carbohydrate specificity, additional experiments should be evaluated to support the idea of protein-carbohydrate interaction. Firstly, bioaccessibility and intestinal permeability of rTBL in the presence of specific glycans. The presence of specific sugars should change the ratio of NDF and DF, because sugars will block the interaction with in intestine cells leading to decreased permeability. Secondly, immunohistochemical analysis in the presence of specific carbohydrates will support the idea of protein-carbohydrate interaction.

Thank you very much. We will take you suggestion for additional experiments since we are now working in some other assays using colon tissue. We will add your suggestion as perspectives in the article. Text was added in page 17, lines 605-610.

  • I think it will be benefit if authors present data on comparison of TBLF and rTBL properties especially in view of cytotoxic effects on cancer cell lines.

We added additional information in page 2, lines 64-83.

  • In my opinion, it will be more harmonic to move data and figures 9 and 10 about antibody design and gastrointestinal digestion from “Materials and Methods” section to “Results” section.

Thank you, Figure 9 was moved to the Results section and now it is Figure 4 in page 5, lines 185-191. Figure 10 is only about a methodology design, so we think that is better to present it in Materials and M section than in the Results section. 

Thank you very much for your comments and suggestions, they were very important for improving the manuscript.

Reviewer 2 Report

Abstract

The abstract should be rewritten. And a brid introduction should be included.

Introduction

The first paragraph of the introduction should be rewriteen since the in vitro and in vivo results are mixed. The authors should first explain all the in vitro results and after that all the in vivo ones.

Moreover, it should be better detailed which is the specific use of this lectins. Is only for colorectal cancer treament? This is not clear.

The following fragment should be at the end of the introduction.

“Due to the low yield, long process, 57 and high costs to obtain the TBLF, the production of a recombinant lectin (rTBL) expressed in yeasts 58 (Pichia pastoris) has been performed in our laboratory [9]. This lectin has structural and functional 59 similarity with one native Tepary bean lectin [10,11], exhibiting similar cytotoxic effects (unpublished 60 data).”

Results

-A brief summary of the main results obtained in the recombinant protein production process should be included. This means: protein yield, protein purity, stability, in vitro protein activity, etc.

-Antibody production results should also be included in the results section. For example Figure 9 is a result and should be included in this section.

-Before adding the everted gut sac was in the solution containing de rTBL, the activity of rTBL should be determined before and after the incubation the addition of all solution components to rTBL to prove pepsin, acidic pH or other components does not affect rTBL protein conformation and activity.

-In Figure 1B and 1C and Figure 3b and 3C controls without rTBL are missing. This is very important to evaluate the impact of rTBL in the intestine.

-Figure 4B: Coud you show the whole western blot?

-Where is this result “Our results showed a decrease of the gut sac weight, 114 indicating basolateral-apical movement” shown?

Materials  & Methods

-4.1. Production of rTBL-1. How protein production is induced? Which inducer is used ? for how long?

Discussion and Conclusions

The discussion should be rewritten including all the results not shown and considering the controls (not included in the figures).

Minor changes:

-line 47: I would add in vitro in this sentence “differential cytotoxic effects in vitro on cancer cell lines”

-line 116: at the end of this sentence (Figure 1C) should be included.

Author Response

Comments and Suggestions for Authors

Abstract

The abstract should be rewritten. And a brid introduction should be included.

Introduction

  • The first paragraph of the introduction should be rewriteen since the in vitro and in vivo results are mixed. The authors should first explain all the in vitro results and after that all the in vivo ones.
  • Moreover, it should be better detailed which is the specific use of this lectins. Is only for colorectal cancer treament? This is not clear.

We have added information in page 2, lines 53-56.

Results

  • A brief summary of the main results obtained in the recombinant protein production process should be included. This means: protein yield, protein purity, stability, in vitro protein activity, etc.

We have added information in page 2, lines 64-83.

  • Antibody production results should also be included in the results section. For example Figure 9 is a result and should be included in this section

Thank you, Figure 9 was moved to the Results section and now it is Figure 4 in page 5, lines 185-191.

  • Before adding the everted gut sac was in the solution containing de rTBL, the activity of rTBL should be determined before and after the incubation the addition of all solution components to rTBL to prove pepsin, acidic pH or other components does not affect rTBL protein conformation and activity.

We monitored rTBL-1 as protein concentration after each step of the in vitro digestion. The biological activity will be measured in future experiments as cytotoxicity on colon cancer cells because the rTBL-1 lacks of agglutination activity. We will add your suggestion as perspectives in the article. Text was added in page 17, lines 606-611.

  • In Figure 1B and 1C and Figure 3b and 3C controls without rTBL are missing. This is very important to evaluate the impact of rTBL in the intestine.

Figure 1: The assays were done without rTBL-1. In these experiments the objective was only to determine the tissue viability, therefore it was no considered to include the rTBL-1 treatment. Respect to Figures 3A and 3B, our results are presented as the difference between treatments with and without rTBL-1, therefore control values were considered in the final result. We have added information in page 15, lines 495-499.

  • Figure 4B: Could you show the whole western blot?

The whole western blots showed in Figure 5 (Figure 4 before) are: (please see attached document)

We consider to show only the band of interest that shows the 30 kDa protein of the Tepary bean lectin.  

  • Where is this result “Our results showed a decrease of the gut sac weight, 114 indicating basolateral-apical movement” shown?

Our results are shown in Figure 1C where the water flux is reported as the difference (decrease) in the gut sac weight.

Materials & Methods

  • 1. Production of rTBL-1. How protein production is induced? Which inducer is used? for how long?

Thank you very much. In fact, it was a mistake, we did not use any inductor for the production of the recombinant lectin. In other cases, methanol is used as inductor but it is not needed in the method we have developed. We have made the corrections in the text, pages 12 and 13, lines 387-407. 

Discussion and Conclusions

  • The discussion should be rewritten including all the results not shown and considering the controls (not included in the figures).

We revised and improved the discussion and conclusions and all the results are included. In the case of the controls, all of them are considered in the calculations of each parameter.

Minor changes:

  • line 47: I would add in vitro in this sentence “differential cytotoxic effects in vitro on cancer cell lines”

Change was done in page 2, lines 50-51

  • line 116: at the end of this sentence (Figure 1C) should be included.

Change was done in page 3, line 140.

Thank you very much for your comments and suggestions, they were very important for improving the manuscript.
